



# Density correction of NRLMSISE-00 in the middle atmosphere (20–100 km) based on TIMED/SABER density data

Xuan Cheng [1,2], Junfeng Yang [1], Cunying Xiao [1,*], Xiong Hu [1]

[1] National Space Science Center, Chinese Academy of Sciences, Beijing, 100190, China
5 [2] University of Chinese Academy of Sciences, Beijing, 100049, China

*Correspondence to*: Cunying Xiao (xiaocy@nssc.ac.cn)

**Abstract.** This paper describes the density correction of the NRLMSISE-00 using more than 15 years (2002–2016) of TIMED/SABER satellite atmospheric density data from the middle atmosphere (20–100 km). A bias correction factor dataset is established based on the density differences between the TIMED/SABER data and NRLMSISE-00. Seven height nodes are 10 set in the range 20–100 km. The different scale oscillations of the correction factor are separated at each height node, and the spherical harmonic function is used to fit the coefficients of the different timescale oscillations to obtain a spatiotemporal function at each height node. Cubic spline interpolation is used to obtain the correction factor at other heights. The spatiotemporal correction function proposed in this paper achieves a good correction effect on the atmospheric density of the NRLMSISE-00 model. The correction effect becomes more pronounced as the height increases. After correction, the relative 15 error of the model decreased by 40–50% in July, especially at ±40 °N in the 80–100 km region. The atmospheric model corrected by the spatiotemporal function achieves higher accuracy for forecasting the atmospheric density during different geomagnetic activities. During geomagnetic storms, the relative errors in atmospheric density at 100 km, 72 km, and 32 km decrease from 41.21%, 28.56%, and 3.03% to −9.65%, 5.38%, and 1.44%, respectively, after correction. The relative errors in atmospheric density at 100 km, 72 km, and 32 km decrease from 68.95%, 24.98%, and 3.56% to 3.49%, 3.02%, and 1.77%, 20 respectively, during geomagnetic quiet period. The correction effect during geomagnetic quiet period is better than that during geomagnetic storms at a height of 100 km. The subsequent effects of geomagnetic activity will be considered, and the atmospheric density during magnetic storms and quiet periods is corrected separately near 100 km. The ability of the model to characterize the mid-atmosphere (20–100 km) is significantly improved compared with the pre-correction performance. As a result, the corrected NRLMSISE-00 can provide more reliable atmospheric density data for scientific research and engineering 25 fields such as data analysis, instrument design, and aerospace vehicles.

## 1 Introduction

The middle atmosphere (20–100 km) is affected by the lower troposphere. For example, the upload of tropospheric Rossby internal waves causes stratospheric planetary-scale disturbances in winter (Huang et al., 2018; Matsuno, 1970) and the uploading of tropospheric gravity waves to the middle atmosphere (Alexander, 1996). The 20–100 km zone can also be affected



by the thermosphere and the upper atmosphere, such as in the reduction of ozone content caused by particle sedimentation (Rozanov et al., 2012; Semeniuk et al., 2011). The coupling of the upper atmosphere and the lower atmosphere with the middle atmosphere causes complex physical and chemical changes in the middle atmosphere. Neutral density is an important environmental parameter in the middle atmosphere. The atmospheric experience model is an important means of obtaining neutral density. Atmospheric experience models are indispensable in science and engineering, and are widely used in data

analysis and engineering design (Kim et al., 2012; Namgaladze et al., 2006; Park et al., 2008; Qian et al., 2018; Yurasov et al., 2008). The MSIS series, CIRA series, and Jacchia series are commonly used empirical models of the atmosphere. These empirical atmospheric models are capable of characterizing the climate change characteristics of atmospheric density. However, because of the irregular spatial and temporal distribution of the observation data, different detection errors from different devices, and simplifications used to establish the atmospheric model, there will be some differences between the

atmospheric model and the actual atmosphere (Chen et al., 2014; Lathuillère and Menvielle, 2010; Vielberg et al., 2018; Xu et al., 2006b; Zhou et al., 2009). Neutral density is the basic input parameter for aircraft design, and density errors between models and observations are one of the main sources of error in spacecraft orbit determination, orbit prediction, and reentry return point prediction. Reducing the differences between atmospheric models and the real atmosphere is a problem that is continually being addressed. Developments in detection technology and the continuous accumulation of data provide an

important foundation for the verification, correction, and improvement of current models.

Various correction studies have considered atmospheric empirical models using observations of density data. For example, the MSIS series of models continues to enrich its database by constantly updating the data source and recalculating the model coefficients using the latest data. Satellite resistance data, accelerometer data, and 95–130 km incoherent scatter radar data were added to the latest version, NRLMSISE-00 (M. Picone et al., 2002). In addition, using the difference between measured

data and the model output to establish the spatiotemporal function of the model correction factor is an effective method for correcting empirical atmospheric models. To modify the JR-71 model, Bergstrom et al. (2002) proposed a linear correction term for the LEO orbital atmospheric density model based on observation data. This method improved the accuracy of the model. Chen et al. (2013) used GRACE and CHAMP satellite density data to correct NRLMSISE-00. The correction effect was verified by a three-day short-term forecast test. The results showed that the corrected model prediction error had been

reduced by more than 50%, significantly improving the prediction accuracy of the model for atmospheric density. Zhou et al. (2009) obtained the relationship from the thermal mass density to the Joule heating power and the high-resolution loop current index by statistically analyzing thermal-layer atmospheric density data from the CHAMP satellite accelerometer during a magnetic storm. Modifying NRLMSISE-00 using an empirical relationship enables better predictions of the atmospheric density during a magnetic storm. Shi et al. (2015) calculated the ratio of the atmospheric density of the two line elements (TLE)

inversion to that of NRLMSISE-00 based on 36 LEO satellite TLE data points from 2000–2002, and calibrated the error in the NRLMSISE-00 output. The results show that the relative root mean square error of NRLMSISE-00 had decreased by approximately 9% after calibration. Mehta et al. (2017) proposed a method for building a semi-physical model driven by thermal layer data using orthogonal decomposition. NRLMSISE-00 was calibrated using GRACE and CHAMP density data





to obtain a more accurate atmospheric density. Zhang et al. (2018) modified the Jacchia–Roberts empirical atmospheric model

using a correction method based on empirical orthogonal function decomposition. This resulted in the 30-day average relative deviation of atmospheric density decreasing by 9.06%. The HASDM model is a modification of Jacchia based on 75 satellite orbit data. The correction method relies on precise orbit determinations, and is costly. Numerous studies on calibrating the density of atmospheric models have used neural network techniques. Perez and Bevilacqua (2015) used the densities from DTM-2013, NRLMSISE-00, and JB2008 as the neural network targets, with CHAMP and GRACE satellite data used for

training, verification, and testing. The resulting density error was better than that before correction. The above studies considered altitudes in the thermosphere, namely the satellite orbital heights. For example, the orbital height of GRACE is ~500 km, and that of CHAMP is ~454 km.

Although there have been some studies on atmospheric model correction methods, they mainly focus on the thermalsphere, particularly the satellite orbital height. There have been no relevant reports on calibrating empirical atmospheric models at

heights of 20–100 km. In this paper, NRLMSISE-00 is used as the modified target model. We construct a spatiotemporal correction function of the model density at 20–100 km for the first time. To evaluate the correction results, statistical methods are used to compare the difference between the atmospheric model and the observed data before and after correction over the period 2002–2016. Observation data from 2017 are used to evaluate the correction effect of the spatiotemporal correction function on the atmospheric density during geomagnetic storms and geomagnetic calm periods. The improved accuracy of the

empirical atmospheric model provides more reliable data support for scientific research and engineering fields such as data analysis, numerical simulation, instrument design, and aircraft design.

## 2 Data source and methods

### 2.1 Database

Thermosphere Ionosphere Mesosphere Energetics and Dynamics (TIMED) is the first solar exploration mission in the NASA

Solar Linkage Program. TIMED was launched on December 7, 2001, and has been accumulating data for over 17 years. The satellite orbit is a sun-synchronous orbit at a height of about 625 km and an orbital inclination of 74.1 °. The period of the satellite orbiting the earth is about 1.6 h. Sounding of the Atmosphere Using Broadband Emission Radiometry (SABER) uses the edge detection method to detect infrared radiation from $CO_2$ in the atmosphere and invert parameters such as atmospheric temperature and density. These data are used to understand the energy exchange and kinetic processes of the intermediate layer,

low thermal layer, and low ionosphere, mesosphere and lower thermosphere (MLT) region. They are also useful for determining basic pressure, temperature, and wind field characteristics due to energy input and output (Mertens et al., 2009; Russell et al., 1999). Satellite precession is slow, taking about 60 days to complete 24 h coverage of local time. Processed raw density data are meshed in the height direction by quality control steps such as preprocessing, information range checking, extreme value checking, and vertical consistency checking (Xiao et al., 2016; Xiao et al., 2017). The grid resolution is 1 km

from 20–100 km. In the NRLMSISE-00 model, density is calculated under the same latitude, longitude, local time,





geomagnetic activity, and solar activity as the satellite trace. The model values are meshed in the height direction in the same way as the observation data.

## 2.2 Correction method

Define the relative error in density between NRLMSISE-00 and TIMED/SABER as:


$$\delta(\%) = \frac{\rho_M - \rho(h)}{\rho(h)} \tag{1}$$

Where $\rho_M(h)$ is the model density and $\rho(h)$ is the observed density.

Define the correction factor $R$ as:

$$R = \frac{\rho(h)}{\rho_M(h)} \tag{2}$$

$$\delta(\%) = \frac{1}{R} - 1 \tag{3}$$

The correction factor $R$ is directly related to the relative error of NRLMSISE-00. We use the gridded observation data and the model to calculate $R$ using Eq. (2). Referring to the modeling method of NRLMSISE-00 below the thermosphere, we set seven height nodes in the range 20–100 km (at 100 km, 90 km, 72 km, 55 km, 45 km, 32 km, and 20 km). At heights of 100 km, 90 km, and 72 km, considering the errors caused by inaccuracies in the tidal wave and traveling planetary wave representations of atmospheric models, spatiotemporal correction functions are established. At the other four height nodes, the errors caused

by inaccuracies in the planetary wave representation of the atmospheric model are considered, and spatiotemporal correction functions are established at the remaining height nodes. Cubic spline interpolation is used to calculate the correction factor at other heights. The correction factor for each height node is meshed with a horizontal resolution of $4° \times 5°$ and a time resolution of 1 h. The data are divided into latitude, longitude, and local time of the 120-day window centered on the day that meshing occurred. In theory, it takes 60 days for the satellite data to cover the global 24 h of local time. Satellite observation data cannot

cover the 24 h of local time at high latitudes in this 60-day window because of the satellite adjustment attitude. Therefore, the meshing of $R$ uses a 120-day sliding window with 1-day steps.

Equation (4) is used to separate 8-h, 12-h, 24-h, 2-day, 6-day, 10-day, 16-day, and 24-day oscillations of $R$ in each horizontal grid at 100 km, 90 km, and 72 km (Pancheva and Mukhtarov, 2011; Xu et al., 2006a).

$$R = R_1 + \sum_{i=1}^{5} \left( R_{1i} \cos\left(\frac{2\pi}{T_i} tloc\right) + R_{2i} \sin\left(\frac{2\pi}{T_i} tloc\right) \right) + \sum_{j=1}^{3} \left( R_{1j} \cos\left(\frac{2\pi}{T_j} tloc\right) + R_{2j} \sin\left(\frac{2\pi}{T_j} tloc\right) \right) \tag{4}$$


Where $R_1$ is the correction factor average, $R_i = \sqrt{R_{1i}^2 + R_{2i}^2}$ and $\varphi_i = \frac{2\pi}{T_i} \tan^{-1}\left(\frac{R_{2i}}{R_{1i}}\right)$ $(i = 1{\sim}5)$ represent the amplitude and phase of components with periods of 2, 6, 10, 16, and 24 days, respectively; $R_j = \sqrt{R_{1j}^2 + R_{2j}^2}$ and $\varphi_j = \frac{2\pi j}{24} \tan^{-1}\left(\frac{R_{2j}}{R_{1j}}\right)$ $(j = 1{\sim}3)$ represent the amplitude and phase of components with periods of 24, 12, and 8 h, respectively.

Further, the annual and semi-annual changes in the above timescale components of the correction factor are separated.





The oscillation components of the annual, semi-annual, quasi-biennial, and 11-year variations of the average term $R_1$ are
       separated. The spherical time harmonic function is fitted to the different timescale component datasets to obtain a modified
       function coefficient set.

       For the other height nodes, the zonal mean of $R$ is calculated, and the annual, semi-annual, quasi-biennial, and 11-year
       variations of the correction factor are fitted in each latitude. A fourth-order Fourier function is used to fit the zonal variation

of different timescale components to obtain different timescale component coefficient sets of the correction function. The
       residual dataset is obtained by subtracting the long-term variation component from the horizontal grid data. The spherical
       harmonic function is fitted to the residual dataset of each height node to obtain a coefficient set of the smaller timescale
       components of the correction function.

## 2.3 Method of assessment

The model density and the observed density from 2002–2016 were meshed before and after correction (grid resolution $2.5^\circ \times$
       $2.5^\circ$). The relative error of the model density before and after correction relative to the observed density was then calculated.
       The relative error of the multi-year average density value of the model before and after correction relative to the multi-year
       observed average was also calculated, and then the difference before and after correction was compared.

       To test the effect of the correction function during different types of geomagnetic activity, the uncorrected observation data

from 2017 were selected to evaluate the correction effect of the atmospheric model under different geomagnetic conditions.
       Observation data from a geomagnetic quiet period and a magnetic storm were selected, and the average relative error and
       standard deviation of the atmospheric model density before and after correction were calculated. The forecasting effect of the
       atmospheric model before and after correction was then compared.

## 3 Results

### 3.1 Difference between model and observations

       There are many studies on the characteristics of atmospheric density variations based on observation data. The atmospheric
       density has characteristic variations in both the vertical and horizontal directions. In the vertical direction, the atmospheric
       density decreases exponentially with height, and seasonal variations are significant. At a fixed height, the density of the summer
       hemisphere is greater than that of the winter hemisphere. The density in the middle and low latitudes is higher than that in the

high latitudes in the spring and autumn. In the stratosphere, planetary wave 1 and wave 2 structures exist at mid-high latitudes
       in the northern hemisphere in January, and the planetary wave 1 structure exists in the southern hemisphere. There is no
       obvious planetary wave structure in the two hemispheres in July. The horizontal structure of the density in the mesosphere in
       winter and summer is similar to that in the stratosphere. The main difference in the distribution is between that in spring and
       that in autumn. The density at the equator is lower than that at mid-latitudes at the height of the stratosphere, whereas it is

higher at the height of the mesosphere. In the thermosphere, the density attains a maximum value over the Antarctic in January





and over the Arctic in July. In spring and autumn, the equator density is higher than that in the northern and southern hemispheres. Perturbations in density can be used to characterize the relative intensity of atmospheric fluctuations. These perturbations in atmospheric density also exhibit seasonal variations. The relative perturbation in atmospheric density increases with height in the vertical direction. The density perturbation in the northern hemisphere is higher than that in the southern

hemisphere in January, and the perturbation in density near 60 °latitude in the southern hemisphere reaches its maximum value in July. During the transition periods of spring and autumn, the density perturbation at the equator is lower than that in the northern and southern hemispheres.

Figure 1 compares the NRLMSISE-00 output with observations. Figure 1(a) and Figure 1(b) show the zonal mean density of the observations and the model at 90 km over the period 2002–2016. Both the observations and the model results exhibit

significant seasonal variations. At low latitudes, the density is high from November to February and low from May to August. At mid-high latitudes, the density of the summer hemisphere is greater than that of the winter hemisphere. The density calculated by NRLMSISE-00 is greater than the observed density, and this is more obvious at middle and low latitudes. Figure 1(c) clearly shows the difference between the model and observations at 90 km. In the vicinity of 60°N and −30°N, the relative error of the model attains maximum values of 68% and 60%, respectively, in June–July. In the vicinity of −60°N and 30°N,

the maximum relative error of the model is 60% and 56%, respectively in December. As shown in Figure 1(d), the relative error in model density increases with height for the same latitude. The relative error at middle and low latitudes is higher than that at high latitudes from 80–100 km. The relative error is mainly around 50%, with the maximum reaching 60%. From 45–80 km, the relative error of the model at middle and low latitudes in the northern hemisphere is greater than that at high latitudes in the northern hemisphere. In contrast, the relative error of the model at middle and low latitudes in the southern hemisphere

is less than that at high latitudes in the southern hemisphere. Below ~45 km, the relative error of the model is generally less than 10%.



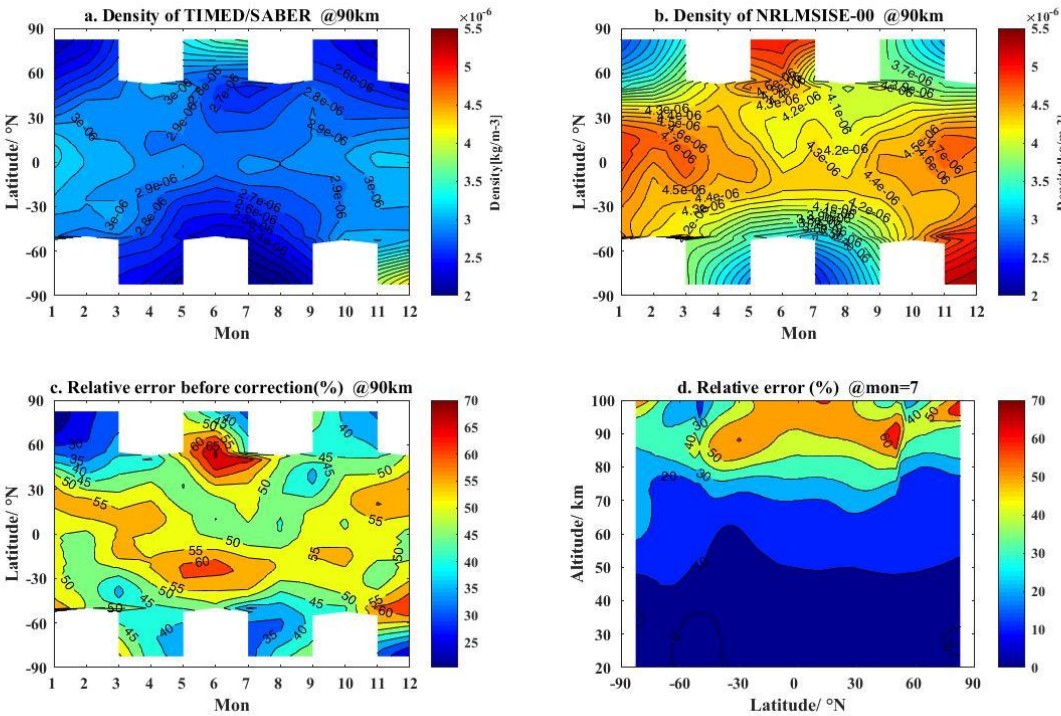

**Figure 1: Zonal mean density of observations and model output from 2002–2016. a. TIMED/SABER at 90 km; b. NRLMSISE-00 at 90 km; c. Time–latitude cross-section of relative deviations in model at 90 km; d. Latitude–height cross-section of relative deviations in model in July.**

## 3.2 Statistical correction results

Figure 2(a) shows the time–latitude cross-section of the zonal mean relative error in the calibrated model at 90 km. The relative error of the calibrated atmospheric model has a maximum value of 15% near 60 °N in June and July. Compared with the relative error of the model before correction, the the relative error of calibrated model reduces the maximum from 60% to 19% in the vicinity of 60 °N from June to July. The maximum error decreases from 60% to 5% near −30°N. Figure 2(b) shows the latitude–height cross-section of relative deviations in the calibrated model in July. The maxima occur at ±40 °N around 80–90 km, representing relative errors of 14% and 17%. From 20–70 km, the relative error of the calibrated model is small. Compared with Figure 1(d), it can be seen that the relative error between the calibrated data and the observations has been significantly reduced, especially in the middle and low latitudes at heights of 80–100 km.



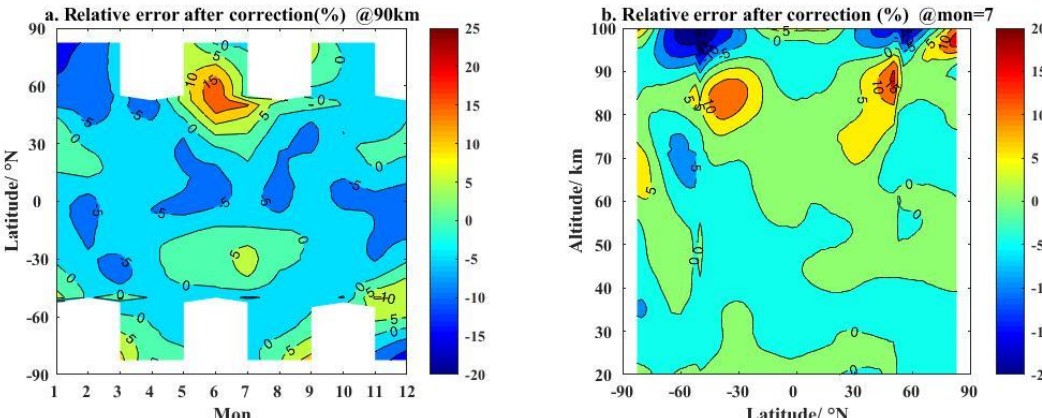

**Figure 2: Zonal mean relative deviations from 2002–2016 after correction. a. Time–latitude cross-section of relative deviations in calibrated model; b. Latitude–height cross-section of relative deviations in calibrated model.**

**3.3 Correction results under different geomagnetic conditions**

Density data from days with an Ap index greater than 80 and less than 27 were selected to calculate the correction effect during a geomagnetic storm and a geomagnetic quiet period, respectively. A large magnetic storm with an Ap index of 106 occurred on September 8, 2017 (day 251), and this was used as the object for evaluating the model during a geomagnetic storm. Most of the days in 2017 were in the geomagnetic quiet period (Ap < 27). The geomagnetic Ap index measured just 5 on May 9, 2017 (day 129), and remained below 10 a few days either side of this day. Thus, the data from May 9, 2017 were selected to evaluate the model during a geomagnetic calm period.

The correction effects were analyzed at three altitude nodes, namely 100 km (representing the low thermosphere), 72 km (mesosphere), and 32 km (stratosphere). The density observations from SABER were extracted at these node heights on days 129 and 251. At the same time, the density and corrected density of NRLMSISE-00 were calculated under the same conditions. We then compared the forecast results given by the model before and after correction.

Figure 3(a) shows the atmospheric density during the geomagnetic storm at 100 km. There is a large deviation between the atmospheric density calculated by NRLMSISE-00 and that observed by SABER. The corrected model density is closer to the density observed by SABER. The correction effects at 72 km and 32 km, shown in Figure 3(b) and Figure 3(c), are considerable. Table. 1 presents the statistical results for the relative error in the NRLMSISE-00 density before and after correction on day 251 and the average relative error and standard deviation of the corrected model density. During the geomagnetic storm, the average relative error of NRLMSISE-00 before correction is 41.42% and the standard deviation is 32.18%. After correction, the average relative error is −9.65% and the standard deviation is 22.56%. The absolute correction of the model is 31.56%. At 72 km, the average relative error before correction is 28.56% and the standard deviation is 8.92%. After correction, the average relative error is 5.38% and the standard deviation is 6.62%. This represents an absolute correction of 23.18%. At 32 km, the average relative error before correction is 3.03% and the standard deviation is 4.96%. This decreases to an average relative



error of 1.44% and standard deviation of 4.29% after correction, an absolute correction of 1.59%. Thus, the model is more
accurate in characterizing the atmospheric density at these three node heights after error correction.

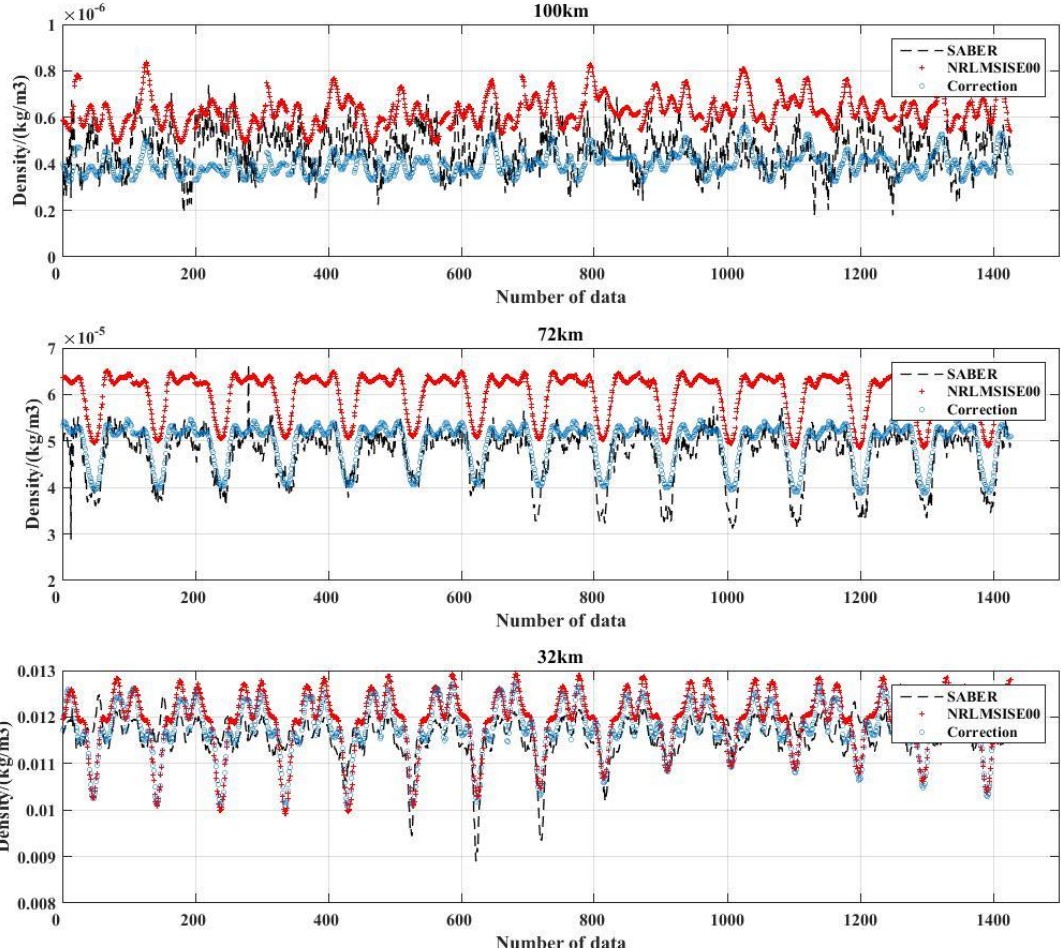

**Figure 3: Density variations at different heights during a geomagnetic storm (day 251 of 2017). a. 100 km; b. 72 km; c. 32 km.**

**Table. 1 Atmospheric density error on satellite orbits at different node on day 251 of 2017 (%)**

|  | 100km | | 72km | | 32km | |
| --- | --- | --- | --- | --- | --- | --- |
|  | **Mean** | **Std** | **Mean** | **Std** | **Mean** | **Std** |
| NRLMSISE00 | 41.21 | 32.18 | 28.56 | 8.92 | 3.03 | 4.96 |





| Correction | -9.65 | 22.56 | 5.38 | 6.62 | 1.44 | 4.29 |
|---|---|---|---|---|---|---|

Figure 4 shows that the density of the three nodes during geomagnetic quiet condition is closer to the SABER observations after correction. The statistical results for the relative error in Table. 2 indicate that the average relative error before correction is 68.95% with a standard deviation of 33.29% at 100 km. After correction, these values drop to 3.49% and 20.65%,

respectively. The absolute correction of the model is 65.46%. Thus, correction significantly improves the accuracy of the density given by NRLMSISE-00 at 100 km in geomagnetic quiet periods. Before correction, the average relative error at 72 km is 24.98% and the standard deviation is 8.04%. The average relative error decreases to 3.02% with a standard deviation of 6.16% after correction. The absolute correction of the model is 21.96%. Thus, error correction of the NRLMSISE-00 model makes a considerable improvement in the accuracy of the atmospheric density at 72 km. At a height of 32 km, the average

relative error before correction is 3.56% and the standard deviation is 1.57%. After correction, the average relative error is 1.77% and the standard deviation of the relative error is 1.91%, an absolute correction of 1.79%. Again, the model accuracy has been improved by the error correction process.

The effect of the correction function varies under the different geomagnetic conditions at around 100 km. After correction, the relative error in the model density decreased from 68.95% to 3.49% during a period of geomagnetic quiet condition. During a

magnetic storm, the relative error in model density decreased from 41.21% to −9.65%. The correction effect during geomagnetic quiet condition is better than that during a geomagnetic storm. In this study, the influence of geomagnetic activity is not considered in the process of establishing the correction function. This factor will be considered in future to improve the correction ability of the spatiotemporal correction function during magnetic storms and to enhance the model's ability to represent the actual atmosphere.





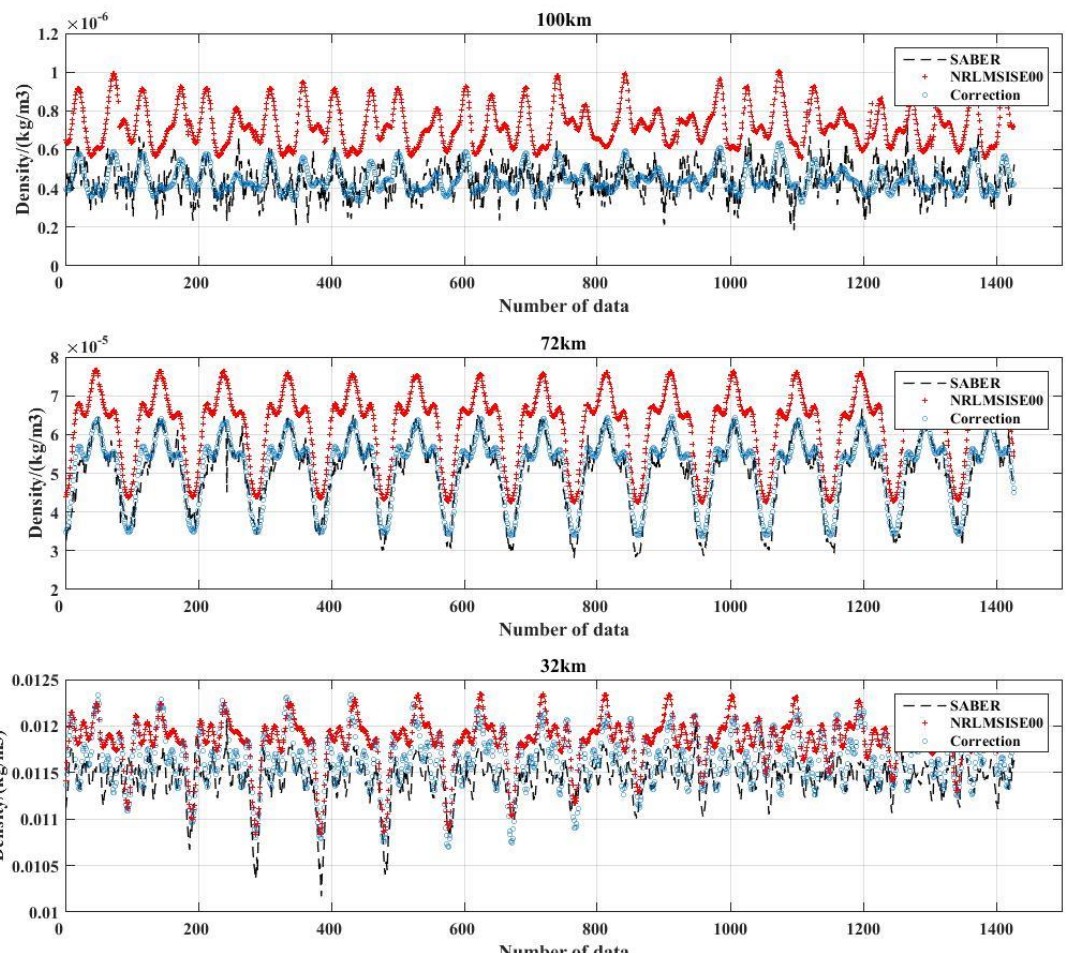


**Figure 4: Same as Figure 3, but for day 129 of 2017.**

**Table. 2 Atmospheric density prediction error on satellite orbits at different node on day 129 of 2017 (%)**

|  | 100km | | 72km | | 32km | |
|---|---|---|---|---|---|---|
|  | **Mean** | **Std** | **Mean** | **Std** | **Mean** | **Std** |
| NRLMSISE00 | 68.95 | 33.29 | 24.98 | 8.04 | 3.56 | 1.57 |
| Correction | 3.49 | 20.65 | 3.02 | 6.16 | 1.77 | 1.91 |





**4 Discussion**

**4.1 Discussion of the correction method**

Considering the error caused by the inaccurate characterization of atmospheric model density as a result of seasonal variations, intra-annual variations, inter-annual variations, and changes in the 11-year cycle of solar activity, we incorporated the above factors into the correction function. In addition, considering the distribution of atmospheric fluctuations in the atmosphere with

height, we divided the range 20–100 km into two height intervals. The first, 70–100 km, contains three height nodes, at 72 km, 90 km, and 100 km, whereas the second, 20–70 km, consists of four height nodes, at 20 km, 32 km, 45 km, and 55 km. To simplify the correction function, inaccuracies in the atmospheric tidal characterization were considered in the 70–100 km height interval. As the NRLMSISE-00 model does not consider traveling planetary waves, we integrated several traveling planetary wave periodic components into the correction function. At a height of 20–70 km, the atmospheric tide is relatively

weak. Below 40 km, the contribution of atmospheric tidal components is small, and the contribution of planetary waves is significant, especially in the winter hemisphere. Therefore, the four height nodes in this interval only consider the atmospheric model to be inaccurate in the representation of planetary waves. To see the correction effect more intuitively, we define the absolute deviation of the relative error between the uncalibrated model and the calibrated model as $\Delta\delta = |\delta_{model}| - |\delta_{correct}|$, where $\Delta\delta$ is the absolute deviation of the relative error, $\delta_{model}$ is the relative error of the model before correction, and

$\delta_{correct}$ is the relative error of the model after correction. $\Delta\delta > 0$ indicates that the model density after correction is closer to that observed by SABER, and the corrective effect is considerable. $\Delta\delta < 0$ indicates that the model density deviates more from the satellite observation of the atmospheric density after correction, i.e., the correction makes the estimate worse. According to Figure 5, the variation in the zonal mean of the model improves in this latitude–altitude cross-section in January, April, July, and October. (Winter and summer are represented by January and July, and spring and autumn are represented by

April and October.) The figure shows that the overall effect of the correction function is significant at 20–100 km, and only the local area model improvement is negative. For example, in the high latitudes of the northern hemisphere in January and the southern hemisphere in April, and the mid-high latitudes of the southern hemisphere in October, the relative error of the corrected model was slightly greater than in the original model. For all areas where the improvement is negative in these months, the results are presented in Table. 3. In the region where $\Delta\delta < 0$ in January, the average value of the improvement

was −0.43% and the standard deviation was 0.41%. In the region where the improvement is negative, the relative error of the modified model is less than 1% compared with that of the model before correction. In areas where the improvement is negative in other months, the average relative error of the model density before and after correction is again less than 1%. Therefore, the spatiotemporal correction function established in this study has a significant effect on the overall correction of the NRLMSISE-00 model.





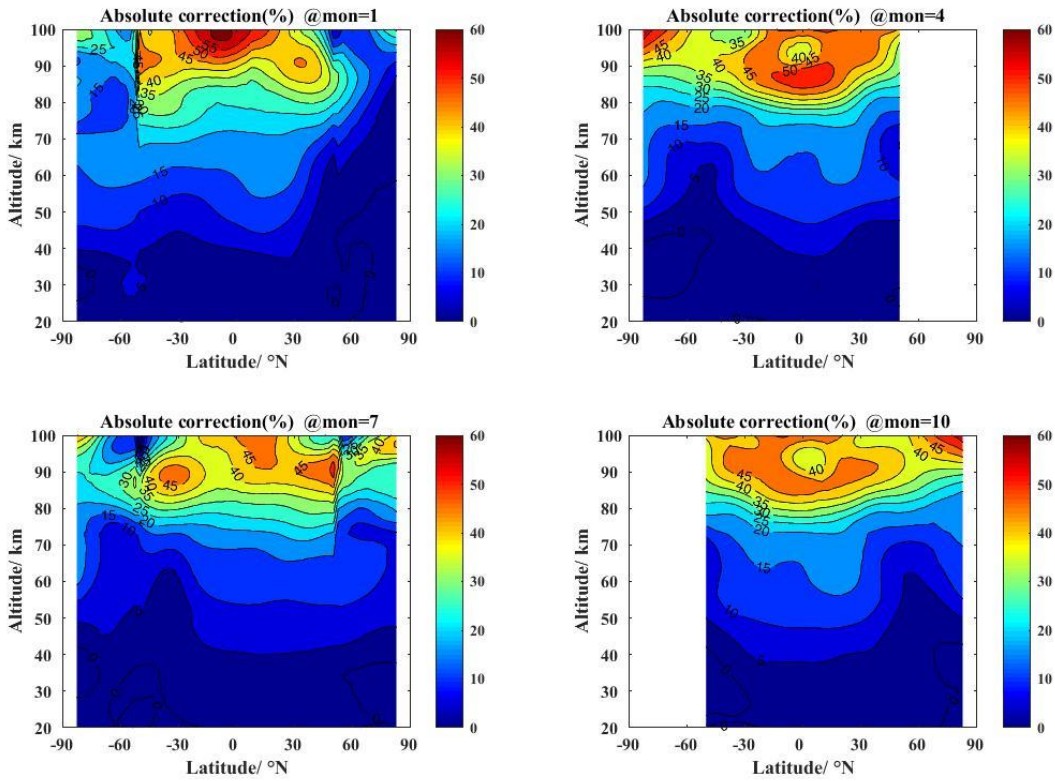


**Figure 5: Latitude-height cross-section of** $\Delta\delta$ **in January, April, July, and October.**

**Table. 3 The average and standard deviations in the area of** $\Delta\delta < 0$ (%)

| January | | April | | July | | October | |
|---|---|---|---|---|---|---|---|
| **Mean** | **Std** | **Mean** | **Std** | **Mean** | **Std** | **Mean** | **Std** 280 |
| -0.43 | 0.41 | -0.47 | 0.32 | -0.81 | 0.90 | -0.44 | 0.34 |

**4.2 Influence of geomagnetic activity**

During geomagnetic storms and substorms, a large number of high-energy particles are injected into the middle and upper atmosphere. There have been many studies on the influence of high-energy particle sedimentation on the middle atmosphere.
The atmosphere is excited and ionized by these high-energy particles, and the chemical composition of the atmosphere changes dramatically in terms of $O_3$, $NO_x$, and $HO_x$ concentrations (Sinnhuber et al., 2012; Zawedde et al., 2016). Changes in these chemical components may cause heating and cooling of the atmosphere, which influences other processes in the middle atmosphere (Kishore Kumar et al., 2018; Ogunjobi et al., 2014; Pancheva et al., 2007). However, there has been little research





on the temperature response of the mesosphere to the sedimentation of high-energy particles. Only a few studies have reported

an increase in temperature at middle and high latitudes during energetic particle precipitation in the mesopause (Savigny et al., 2007; Yuan et al., 2015). There is a strong coupling between the ionosphere and the neutral atmosphere during geomagnetic storms. Both the ionosphere and the lower atmosphere exhibit quasi-two-day waves (QTDWs) in the geomagnetic quiet period because of upward coupling of QTDWs. During a minor sudden stratospheric warming (SSW), the amplitude of the QTDWs in the upper stratosphere is weakened and intensified after the geomagnetic storms. However, the QTDWs in the ionosphere

are enhanced after geomagnetic storms. The possible reason is that the same frequence of interplanetary magnetic field and QTDWs with the modulation of different phase of storm induced circulation and   the climatological circulations in the lower mesosphere might have led to enhancement and inhibition of the amplitude of the QTDWs over different hemisphere (Mengistu Tsidu and Abraha, 2014). Other studies have shown that the atmospheric density in the mesosphere over polar regions is negatively correlated with geomagnetic activity (Yi et al., 2018; Yi et al., 2017). The mechanism by which the atmospheric

density in the mesosphere over polar zones responds to geomagnetic activity is considered to be different from that in the thermosphere (Lei et al., 2008; Xu et al., 2015). The results of these studies indicate that the response mechanism of mid-atmospheric density to geomagnetic activity requires further clarification and exploration to promote the establishment of new models and improve existing models.

## 4.3 Influence of solar activity

Solar activity is another important factor in correction factor modeling. From the modeling of temperature, NRLMSISE-00 calculates the density of each component using the physical relationship between the temperature, static equilibrium, and ideal gas state equation. The sum of the density in each of the components is the total density. The response of the NRLMSISE-00 atmospheric density to solar activity depends, to some extent, on the temperature response to solar activity. The sensitivity to solar activity varies with altitude, with studies showing that the effects of solar activity decrease with height. The absorption

of solar radiation by ozone in the stratosphere plays an important part in the energy cycle and kinetics of the heating and cooling of the stratosphere. The changes are influenced by kinetics, chemistry, and other parameters (Staehelin et al., 2001), while the 11-year solar cycle has little effect on the total ozone content (L. Hood, 1997). There are conflicting results regarding the response of MLT to solar activity. Luebken (2001) used sounding data to study the structural characteristics of the mesosphere over the polar regions in the past 35 years. The results indicate that the temperature structure of the mesosphere

in this zone has not changed significantly. Detailed analysis showed that the solar cycle has little effect on temperature. However, Remsberg and Deaver (2005) used HALOE to examine temperature changes in the upper and middle stratosphere from 1991–2004 temperature data. In the upper and middle part of the tropical stratosphere, the temperature responded significantly to the solar activity week. In the upper part of the tropical stratosphere and the subtropical mesosphere, the trend suggested a linear decline, but this phenomenon has not been found in the tropical mesosphere. Therefore, the mechanism of

the influence of solar activity on the middle atmosphere requires further study so that atmospheric models can better characterize the influence of solar activity on the middle atmosphere.



## 5 Conclusions

In this study, we used density data from TIMED/SABER for the period 2002–2016 to correct the density of the empirical atmospheric model NRLMSISE-00 at a height of 20–100 km for the first time. By analyzing the difference between the model
output and the observations, a method for establishing a spatiotemporal correction function for NRLMSISE-00 was proposed. According to the spatiotemporal distribution characteristics of the correction factor dataset, different timescale oscillations in the correction factors appear at every node. The spherical harmonic function was used to fit the coefficients of the separated components to obtain the spatiotemporal correction function for NRLMSISE-00 over the range 20–100 km. The corrected model density was calculated using this function and the results were evaluated. To this end, the main conclusions from this
study are as follows:

(1) The model and observations exhibit the same variation in the horizontal and vertical directions, but there is a certain deviation. The relative error of the model at middle and high latitudes is greater in the summer hemisphere than in the winter hemisphere. At 60 N, the relative error of the model can reach 60% at a height of 90 km from June to July. The relative error of the model increases with height in the vertical direction, especially in the region of 80–100 km.

(2) The accuracy of the calibrated model is better than that of NRLMSISE-00. The accuracy of the model is significantly improved at altitudes of 80–100 km. The maximum value of the relative error at latitudes of ±40 N decreased by about 40–50% in July. There was also a significant correction effect in other months. At 90 km, the accuracy of the model further decreased by about 55% at −30°N and by 54% at 60°N from June to July.

(3) The correction function produces a significant improvement in the prediction of atmospheric model density under different
geomagnetic conditions. After correction, the relative errors in model density at 100 km, 72 km, and 32 km decreased from 41.21%, 28.56%, and 3.03% to −9.65%, 5.38%, and 1.44%, respectively, during an geomagnetic storm. During a geomagnetic quiet period, the relative errors in model density at 100 km, 72 km, and 32 km decreased from 68.95%, 24.98%, and 3.56% to 3.49%, 3.02%, and 1.77%, respectively. In the low thermosphere, the correction effect of the function in the geomagnetic calm period is significantly better than that in the magnetic storm period. Subsequent work will consider the effects of geomagnetic
activity and optimize the ability of the spatiotemporal correction function to correct the atmospheric density during magnetic storms.

The density mechanism in response to solar activity and geomagnetic activity requires further investigation in the range 20–100 km. This theoretical study provides a technical basis for the establishment of new models and the improvement of existing models, and enhances the ability of NRLMSISE-00 to represent the real atmosphere. By correcting the density over the range
20–100 km for NRLMSISE-00, higher-quality initial and background fields can be provided for numerical simulations and predictions in scientific research. Additionally, reliable atmospheric density data can be derived for aircraft design, simulation, and flight tests in aerospace and other engineering fields.





**Author contribution**

Xuan Cheng and Cunying Xiao have made contributions to the conception and design of this work. Xuan Cheng and Junfeng

Yang have made contributions to the acquisition, analysis and interpretation of data and creation 355 of spatiotemporal

correction function. Xuan Cheng have drafted the manuscript and Cunying Xiao and Xiong Hu have revised it critically for

important intellectual content.

**Author contribution**

The authors declare that they have no conflict of interest.

**Acknowledgements**

This study was supported by the National Key Research and Development Program of China (2016YFB0501503), the Strategic

Priority Research Program of Chinese Academy of Sciences Grant No. XDA17010301, the National Natural Science

Foundation of China (Grant No. 11872128). We thank the TIMED/SABER working group (http://saber.gats-

inc.com/data_services.php) to provide the observation data and the Space Environment Forecast Center of the National Space

Science Center of the Chinese Academy of Sciences to provide the F10.7 and Ap index.

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
