# Peer review of "Density correction of NRLMSISE-00 in the middle atmosphere (20–100 km) based on TIMED/SABER density data"

_Annales Geophysicae, 2019_

## Referee Comment (RC1) · Anonymous Referee #1 · 11 Sep 2019

The reviewer deems this study is innovative and meaningful. However, some questions should be clarified in the paper before its publication, as following,

1. Line 87: earth -> Earth;

2. Line 175: Below $\sim$45 km -> Below about 45 km;

3. Line 213$\sim$214: This decreases to an average relative error of 1.44% and standard deviation of 4.29% after correction, an absolute correction of 1.59%. -> These decrease to an average relative error of 1.44% and standard deviation of 4.29% after correction, with an absolute correction of 1.59%. The same mistake appears in line 231. Besides, there are also some other grammar mistakes which will not be listed

[Figure]

here, please correct them in the manuscript. 4. The value "68%" in line 169 are not consistent with "60%" in line 184, please check it. The same for "15%" in line 183 and "19%" in line 184.

5. Why the deviations are so large at 72 km given by NRLMSISE-00 in Fig. 3b and Fig. 4b? An explanation is suggested in the manuscript.

6. It is found that the average relative error decreases a lot while the refinement of standard deviation is not so clear, how to explain it? Especially the case of 32 km in Table 1 and Table 2.

7. The average deviation becomes very small while the standard deviation decreases slightly, how to explain it?

8. Throughout the paper, the quantitative results after corrections are insufficient. In order to make a comprehensive assessment, more quantitative results under different conditions (latitude, month, altitude, local time) should be provided.

9. In line 111, it is mentioned that "Cubic spline interpolation is used to calculate the correction factor at other heights". However, there is no correction result at other heights. This is very important for the assessment of the correction method. Hence some correction results at non-node heights are suggested to be presented in the paper.

---

## Author Comment (AC1) · 19 Oct 2019

I indeed appreciate the reviewer for very useful and constructive comments and suggestions. We have studied comments carefully and have made correction which we hope meet with approval. Revised portions are marked in blue in the paper and detailed information of changes in the supplement. The main corrections in the paper and the responds to the reviewer's comments are as flowing:

Comment 1: Line 87: earth -> Earth;

Reply 1: In line 87, the statement of "earth" was corrected as "Earth".

[Figure]

Comment 2: Line 175: Below âĹij45 km -> Below about 45 km;

Reply 2: In line 179, the statement of "Below âĹij45 km" was corrected as "Below about 45 km". In line 71~72, the statements of "the orbital height of GRACE is ~ 500 km, and that of CHAMP is ~ 454 km" were corrected as "the orbital height of GRACE is about 500 km, and that of CHAMP is about 454 km".

Comment 3: Line 213âĹij214: This decreases to an average relative error of 1.44% and standard deviation of 4.29% after correction, an absolute correction of 1.59%. -> These decrease to an average relative error of 1.44% and standard deviation of 4.29% after correction, with an absolute correction of 1.59%. The same mistake appears in line 231. Besides, there are also some other grammar mistakes which will not be listed here, please correct them in the manuscript.

Reply 3: In line 253~254 and line 270~271, these grammer mistakes have been corrected.

Comment 4: The value "68%" in line 169 are not consistent with "60%" in line 184, please check it. The same for "15%" in line 183 and "19%" in line 184.

Reply 4: The value "68%" in line 164 and line 195 were checked and revised consistently. The value "19%" in line 193 and line 195 were checked and revised.

Comment 5: Why the deviations are so large at 72 km given by NRLMSISE-00 in Fig. 3b and Fig. 4b? An explanation is suggested in the manuscript.

Reply 5: As Reviewer suggested that an explanation was added in line 180~187 as "Atmospheric planetary waves, atmospheric tidal waves and atmospheric gravity waves are important sources of atmospheric disturbances above 70 km (Pancheva and Mukhtarov, 2011, Xiao, et al., 2016, Zhang, et al., 2006). The atmospheric density of the NRLMSISE-00 model is calculated from the atmospheric temperature. In the UMLT region, there is a large error between the model temperature and the TIMED/SABER observation (Xuan, et al., 2018). The contribution of traveling planetary waves to atmospheric disturbances and the inaccurate estimation of atmospheric tides are the possible reasons for the large error of atmospheric temperature in the UMLT region. The atmospheric model transmits the error caused by the inaccurate representation of the atmospheric temperature disturbance to the atmospheric density, which makes the atmospheric density have larger errors in the UMLT region."

Comment 6: It is found that the average relative error decreases a lot while the refinement of standard deviation is not so clear, how to explain it? Especially the case of 32 km in Table 1 and Table 2.

Reply 6: Because the relative error of NRLMSISE-00 density increases with height, the relative error is smaller at lower heights (the correction factor is close to 1). The correction effect of the spatiotemporal correction function on the model is not significant at lower heights, so the change of the standard deviation is not clear at 32 km.

Comment 7: The average deviation becomes very small while the standard deviation decreases slightly, how to explain it?

Reply 7: The statistical results of the relative error of the model before and after correction given in Tables 1 and 2 include the correction results of different latitude and longitude under the satellite orbits. From figure 1-4, the relative error of the model is different with the distribution of latitude and longitude before or after correction. The statistical results of the average relative error before and after correction directly reflect the overall correction effect of the model at the observation points under different latitude and longitude. The standard deviation reflects the dispersion of the relative error at different latitude and longitude, indicating the difference of correction effect under different latitude and longitude. The standard deviation decreases slightly may be related to the modeling method of this paper and the property of the satellite data used. Since the coverage of the TIMED/SABER satellite alternates every 60 days, the observation capability is limited at high latitudes above $\pm52°$, resulting in limited to data volume. Taking this factor into consideration, this paper uses a 120-day window to

mesh the correction factor R so that the satellite data can cover both the high and the latitudes of the northern and southern hemispheres in the process of establishing the spatiotemporal correction function. These may be the reasons for the limited correction capability of the space-time correction function in high latitudes, and the standard deviation decreases slightly.

Comment 8: Throughout the paper, the quantitative results after corrections are insufficient. In order to make a comprehensive assessment, more quantitative results under different conditions (latitude, month, altitude, local time) should be provided.

Reply 8: More quantitative results were provided in section 3.1 In order to make a comprehensive assessment, the corresponding changes in the manuscript are as follows: (1) Latitude–month cross-section of relative error (figure 3) and latitude-height cross-section of relative error (figure 4) were added in section 3.2. The latitude–month cross-section of relative error after correction at 90 km, 60 km and 30 km were added in figure 3. The latitude-height cross-section of relative error after correction in January, April, and October were added in figure 4. And some detailed analysis of the correction effect was added in the manuscript. (2) In order to make it more intuitive to compare with the corrected results in (1), the density of observations and model output at 90 km in figure 1 were delete in section 3.1 and the latitude–month cross-section of relative error before correction at 60 km and 30 km were added in figure 1. Besides, the latitude-height cross-section of relative error before correction in January, April, July, and October were added in figure 2. And some detailed analysis was added in the manuscript. (3) The relative error of the model varies with local time before and after correction was added in section 3.3. (4) Adjust the original section 3.3 to section 3.4. (5) In line 239, "(both node heights and non-node heights were contained)" was added.

Comment 9: In line 111, it is mentioned that "Cubic spline interpolation is used to calculate the correction factor at other heights". However, there is no correction result at other heights. This is very important for the assessment of the correction method. Hence some correction results at non-node heights are suggested to be presented in
the paper.

Reply 9: The statistical correction results given in Section 3.2 include the results of non-node heights. For example, the latitude–month cross-section of relative error after correction at 30 km and 60 km shown in figure 3 are the corrected results for the non-height nodes, and the latitude–month cross-section of relative error at 90 km is the correction result for the height nodes. The data used in Figure 4 have a height resolution of 1 km and contain correction results for 7 node heights and 74 non-node heights. In Section 3.4, the result of 72 km in Figures 6 and 7 were replaced with that of 70 km in order to include a non-node height in the correction results under different geomagnetic activity conditions.

Please also note the supplement to this comment:
https://www.ann-geophys-discuss.net/angeo-2019-93/angeo-2019-93-AC1-supplement.pdf

―――――――――――――――――――

**Supplement:**

[revised manuscript text omitted]
. At 60 km, the relative error of atmospheric model at low and medium latitudes is mainly around 15%. From December to January, the maximum value of the relative error of atmospheric model in low latitudes can reach 23%. At 30 km, the relative error maximum value is 7% in the low latitude area from January to April. In the mid-latitudes of the southern hemisphere, there is a minimum value near August, with a relative error of -5%.

[Figure]

**Figure 1: Latitude–month cross-section of relative error before correction. a. at 90 km; b. at 60 km; c. at 30 km.**

As shown in Figure 2, the relative error in model density increases with height for the same latitude. The relative error at middle and low latitudes is higher than that at high latitudes from 80–100 km. Near the equator in January, the maximum relative error of the model reaches 79% at 100 km. From 45–80 km, the relative error of the model in the northern hemisphere is greater than that in the northern hemisphere. In July, the relative error is mainly around 50%, with the maximum reaching 68% at 80-100 km. From 45–80 km, the relative error of the model at middle and low latitudes in the northern hemisphere is greater than that at high latitudes in the northern hemisphere. In contrast, the relative error of the model at middle and low latitudes in the southern hemisphere is less than that at high latitudes in the southern hemisphere. Below about 45 km, the relative error of the model is generally less than 10%. Atmospheric planetary waves, atmospheric tidal waves and atmospheric gravity waves are important sources of atmospheric disturbances above 70 km (Pancheva and Mukhtarov, 2011, Xiao, Hu, Wang and Yang, 2016, Zhang, et al., 2006). The atmospheric density of the NRLMSISE-00 model is calculated from the atmospheric temperature. In the UMLT region, there is a large error between the model temperature and the TIMED/SABER observation (Xuan, et al., 2018). The contribution of traveling planetary waves to atmospheric disturbances and the inaccurate

185  estimation of atmospheric tides are the possible reasons for the large error of atmospheric temperature in the UMLT region. The atmospheric model transmits the error caused by the inaccurate representation of the atmospheric temperature disturbance to the atmospheric density, which makes the atmospheric density have larger errors in the UMLT region.

[Figure]

Figure 2:Latitude-height cross-section of relative error before correction in January, April, July, and October.

190  **3.2 Statistical correction results**

(1) Latitude–month

Figure 3 shows the latitude–month cross-section of the zonal mean relative error in the calibrated model at different heights. At 90 km, the relative error of the calibrated atmospheric model has a maximum value of 19% near 60 N in June and July. Compared with the relative error of the model before correction, the the relative error of calibrated model reduces the maximum

195  from 68% to 19% in the vicinity of 60 N from June to July. The maximum error decreases from 62% to 7% near −30 N. At 60 km, the relative error of calibrated model reduces the maximum from 23% to less than 4% near equator in December-January after correction. And in the vicinity of -80 N, the the relative error of calibrated model reduces the maximum from

20% to 3% in April and 26% to 4% in August. At 30 km, the the relative error of calibrated model reduces the maximum from 7% to less than 2% in low latitudes in January to April.

200

[Figure]

**Figure 3: Latitude–month cross-section of relative error after correction at 90 km, 60 km and 30 km**

(2) Latitude–altitudes

205 Figure 4 shows the latitude–height cross-section of relative deviations in the calibrated model in January, April, July, and October. As it can be seen from the figure that the correction effect of the atmospheric model is significant above 70 km. In January, the the relative error of calibrated model reduces the maximum from 79% to 17% at 100 km near equator. In April, the the relative error of calibrated model reduces the maximum from 64% to 16% around 80 km near equator. In July, the maxima occur at ±40 °N around 80–90 km, representing relative errors of 14% and 17%. In October, the the relative error of

210 calibrated model has a maximum value of 12% around 80 km. From 20–70 km, the relative error of the calibrated model is small. Compared with Figure 2, it can be seen that the relative error between the calibrated data and the observations has been significantly reduced, especially in the middle and low latitudes at heights of 80–100 km.

[Figure]

**Figure 4: Latitude-height cross-section of relative error after correction in January, April, July, and October.**

**3.3 Correction results under different local time**

In order to compare the correction results of different local times (LT), the relative error was calculated at different local times before and after the correction. Figure 5 shows the relative error of the atmospheric model before and after correction for different local time. At 90 km and 60 km, the relative error of the atmospheric model before correction is positive at different geographical locations, indicating that the model value is greater than the satellite observations. At 90 km, the relative error is between 30% and 100% befror correction and the relative error is between ±20% after correction. In the latitude of ±50 °N, the relative error has local maximum at about LT 6 and LT 19. The smallest relative error is seen in the LT range of about 10-14, but there is no more data in that LT range. In the latitude of ±30 °N, the relative error has a local maximum in the LT range of about 2-7, and there is a local minimum in the LT range of 14-18. At 60 km, the relative error is mainly between 5% and 25% befror correction and the relative error is mainly between ±5% after correction.

At 30 km, the relative error of the atmospheric model before correction has both positive and negative values at different local times and the relative error of the model has improved to some extent. At the same height, the relative error of the atmospheric

model varies with local time in the northern hemisphere is similar to that in the southern hemisphere. The relative error varies with local time in a similar sine or cosine function. It can be considered that the relative error has a relationship with the local time, as diurnal waves or semidiurnal waves.

[Figure]

230

**Figure 5: The relative error of the model varies with local time before (blue lines) and after (red lines) correction in 2008.**

[revised manuscript text omitted]

Xuan, C., Cunying, X., Xiong, H., Junfeng, Y., 2018. Evaluation of atmospheric empirical model based on TIMED/SABER satellite temperature data. Sci Sin-Phys Mech Astron 48 (10), 79-93.

Yi, W., Reid, I., Xue, X., J. Murphy, D., M. Hall, C., Tsutsumi, M., Ning, B., Li, G., P. Younger, J., Chen, T., Dou, X., 2018. High- and Middle-Latitude Neutral Mesospheric Density Response to Geomagnetic Storms. Geophysical Research Letters 45 (1), 436-444.

Yi, W., Reid, I., Xue, X., P Younger, J., J Murphy, D., Chen, T., Dou, X., 2017. Response of neutral mesospheric density to geomagnetic forcing. Geophysical Research Letters 44 (16), 8647-8655.

Yuan, T., Zhang, Y., Cai, X., She, C.Y., Paxton, L., 2015. Impacts of CME Induced Geomagnetic Storms on the Mid-latitude Mesosphere and Lower Thermosphere Observed by a Sodium Lidar and TIMED/GUVI. Geophysical Research Letters 42 (18), 7295-7302.

Yurasov, V.S., Nazarenko, A.I., Alfriend, K.T., Cefola, P.J., 2008. Reentry time prediction using atmospheric density corrections. Journal Of Guidance Control And Dynamics 31 (2), 282-289.

Zawedde, A., Nesse Tyssøy, H., Hibbins, R., Espy, P., Ødegaard, L.-K., Sandanger, M.I., Stadsnes, J., 2016. The Impact of Energetic Electron Precipitation on Mesospheric Hydroxyl during a Year of Solar Minimum: EEP IMPACT ON MESOSPHERIC OH DURING SOLAR MINIMUM. Journal of Geophysical Research: Space Physics 121 (6), 5914-5929.

Zhang, H., Gu, D., Duan, X., Wei, C., 2018. Atmospheric density model calibration using emperical orthogonal function. Acta Aeronautica et Astronautica Sinica 39 (S1), 722263.

Zhang, X., Forbes, J.M., Hagan, M.E., Russell, J.M., Palo, S.E., Mertens, C.J., Mlynczak, M.G., 2006. Monthly tidal temperatures 20–120 km from TIMED/SABER. Journal of Geophysical Research 111.

Zhou, Y.L., Ma, S.Y., Lühr, H., Xiong, C., Reigber, C., 2009. An empirical relation to correct storm-time thermospheric mass density modeled by NRLMSISE-00 with CHAMP satellite air drag data. Advances in Space Research 43 (5), 819-828.

---

## Referee Comment (RC2) · Anonymous Referee #2 · 24 Feb 2020

Density correction of NRLMSISE-00 in the middle atmosphere (20–100 km) based on TIMED/SABER density data

by

Xuan Cheng, Junfeng Yang, Cunying Xiao, Xiong Hu

Absolute neutral air density observations at the upper atmosphere are rare and very often empirical models such as NRLMSIS-00 are the only source of information of the neutral air density. 30 years ago, this was also the case in the middle atmosphere, which is the primary atmospheric region of interest in the submitted manuscript. The authors aim to correct the NRLMSIS-00 at altitudes between 20-100km with respect to

the SABER observations onboard TIMED. The correction includes climatological information on planetary waves and tides as well as solar effects to obtain a better model. Further, the authors claim that there is no other information available to account for the density at the middle atmosphere, which accounts for the variability due to atmospheric waves. The manuscript is not acceptable for publication as major statements and a substantial part of the implemented wave dynamics is not correct making the whole manuscript obsolete.

Major concerns: 1) There is no need to improve the NRLMSIS-00 climatology to account for atmospheric waves. There is high-quality density information available throughout the middle atmosphere from meteorological analysis as well as reanalysis data sets (see NAVGEM-HA, MERRA2, ECMWF, etc.). Further, nudged GCM's provide additional information on the wave-driven variability of the neutral air density or long-term changes in the middle atmosphere (WACCM, WACCM-X etc.). Some of these data sets even allow us to resolve the day-to-day variability of the neutral air density variability due to planetary waves and tides as well as gravity waves.

2) Planetary waves and tides show a strong seasonal and inter-seasonal variability. In particular, the phases of atmospheric waves are variable, causing issues using empirical climatologies for a certain atmospheric wave. This variability manifests in the occurrence of sudden stratospheric warmings, which evolve quite different from year to year.

This phase variability is also an important issue for tides, which show a significant response to changes in the middle atmosphere resulting in a interday variability that cannot be covered using 60-day climatological output.

3) Why correct an empirical model, which is still very good, if one has weather models that provide the neutral air density and the associated variability for free?

4) Further, the reviewer is concerned about the claims of the authors that SABER represents a 'true' measure of the neutral air density. At the MLT SABER observes

mostly CO2 and converts it to an absolute neutral density assuming a volume mixing ratio, which shows also rather large errors (see Remsberg et al., 2008 and Rezac et al., 2015). The errors at the MLT of the volume mixing ratio are as high as 15-30%. This also limits the possibility to draw fundamental conclusions about the absolute density scale. Further, it is worth to consider that some a priori information in the SABER retrievals is taken from WACCM (chemical equilibrium codes).
* * *

---

## Author Comment (AC2) · 25 Feb 2020

I thank the reviewers for valuable comments and constructive critique.

Based on your comments, we need to restate the necessity and significance of revising the model. At present, researches on the revised NRLMSISE-00 model is aimed at engineering applications. For example, the accuracy of the atmospheric density of NRLMSISE-00 model cannot meet the normal operation of the satellite. Therefore, the atmospheric density of the model is modified at the orbital altitude of the satellite, at an altitude of 400$\sim$600km. At present, the research on the modification of NRLMSISE-00 density of 20-100km has not been reported, but the modification of the

atmospheric density of the model within this height range is also very necessary. In addition, there is some error in the atmospheric density of NRLMSISE-00 output. More accurate NRLMSISE-00 atmospheric density data are needed when satellites fall into the atmosphere or when space vehicles fly in this altitude range.

Finaly, all comments were carefully considered and addressed. Answers to all the questions are presented below.

(1) There is no need to improve the NRLMSIS-00 climatology to account for atmospheric waves. There is high-quality density information available throughout the middle atmosphere from meteorological analysis as well as reanalysis data sets (see NAVGEM-HA, MERRA2, ECMWF, etc.). Further, nudged GCM's provide additional information on the wave-driven variability of the neutral air density or longterm changes in the middle atmosphere (WACCM, WACCM-X etc.). Some of these data sets even allow us to resolve the day-to-day variability of the neutral air density variability due to planetary waves and tides as well as gravity wavesãĂĆ

Reply: I agree with the reviewer's opinions that reanalysis data can provide high-quality density information of middle atmosphere density and the numerical models (WACCM, waccm-x) can be used for numerical simulation of the variations of atmospheric density waves and long-term changes. Re-analysis data and numerical models are widely used in the field of scientific research, but re-analysis data and numerical models are less used than NRLMSISE-00 empirical models in engineering applications. In engineering applications, NRLMSISE-00 empirical model can simply and quickly output the required atmospheric data, which are why NRLMSISE-00 model is widely used in satellite orbit prediction and other engineering fields. Since the modification the atmospheric density of the NRLMSISE-00 between 20 and 100km has corresponding requirements and application scenarios, the NRLMSISE-00 model needs to be corrected.

(2) Planetary waves and tides show a strong seasonal and inter-seasonal variability. In particular, the phases of atmospheric waves are variable, causing issues using empirical climatologies for a certain atmospheric wave. This variability manifests in the occurrence of sudden stratospheric warmings, which evolve quite different from year to year. This phase variability is also an important issue for tides, which show a significant response to changes in the middle atmosphere resulting in a interday variability that cannot be covered using 60-day climatological output.

Reply: I couldn't agree with you more. The phase of atmospheric wave is variable, which is an important problem in scientific research. TIMED/SABER is limited by its own observation system, which requires 60 days to cover a 24-hour local time. In order to account for the impact of atmospheric tidal waves, the 60-day climatological output had to be considered. The correction of interday variability of atmospheric tidal phase will be added to the correction function in the subsequent study.

(3) Why correct an empirical model, which is still very good, if one has weather models that provide the neutral air density and the associated variability for free?

Reply: Compared with the numerical model, the NRLMSISE-00 model has the advantages of simple use and high calculation efficiency. Compared with other empirical models, the model has higher accuracy and is widely used in the engineering field, even if the model has been released for free for about 20 years. In the application of satellite orbit determination and orbit prediction, many experts have been carrying out research on the modification of NRLMSISE-00 model in order to obtain more accurate atmospheric density data of orbital altitude. At present, this model is typically used in the basic research (such as track prediction, aerodynamic heating, etc.) of space vehicles flying at altitudes of 20-100km. However, it is necessary to improve the accuracy of the atmospheric density data of this model so as to provide higher precision density data for relevant studies. Therefore, we need to explore some correction methods to modify the results of 20-100km to better meet the engineering requirements.

(4) Further, the reviewer is concerned about the claims of the authors that SABER represents a 'true' measure of the neutral air density. At the MLT SABER observes

mostly CO2 and converts it to an absolute neutral density assuming a volume mixing ratio, which shows also rather large errors (see Remsberg et al., 2008 and et al., 2015). The errors at the MLT of the volume mixing ratio are as high as 15-30%. This also limits the possibility to draw fundamental conclusions about the absolute density scale. Further, it is worth to consider that some a priori information in the SABER retrievals is taken from WACCM (chemical equilibrium codes)

Reply: Although the errors of CO2 volume mixing ratio are as high as 15-30% in the MLT region, it can affect the accuracy of neutral air density and the accuracy of correction to a certain extent. Considering the absolute neutral air density observations at the middle atmosphere are rare. There is no alternative to the TIMED/SABER data that can cover the globe and have continuous observations over a long period of time. In addition, according to the results obtained in this paper, the density of NRLMSISE-00 can be effectively corrected to be closer to that of SABER by using the correction method in this paper. Therefore, the correction method in this paper is feasible. After obtaining other high-precision detection data, the method in this paper can be used to further improve the accuracy of NRLMSISE-00.